Optimizing marine vehicles industry: a hybrid analytical hierarchy process and additive ratio assessment approach for evaluating and selecting IoT-based marine vehicles

Ullah Khan Habib 1
Abbas Muhammad 2
Nazir Shah 3
Khan Faheem faheem@gachon.ac.kr 4
Hussain Jamil jamil@sejong.ac.kr 5
1 Accounting & Information Systems at the College of Business and Economics, Qatar University , Doha , Qatar
2 Faculty of Computer Science and Engineering, Ghulam Ishaq Khan Institute of Engineering Sciences and Technology, Pakistan , Swabi , KPK , Pakistan
3 Department of Computer Science, University of Swabi, Pakistan , Swabi , KPK , Pakistan
4 Computer Engineering, Gachon University , Sengnam-si , Seoul , Republic of South Korea
5 Department of Artificial Intelligence (AI) and Data Science, Sejong University , Seoul , Republic of South Korea
Pires Ivan Miguel
Electronic publication date: 2024 Oct 15
Publication date: 2024
Volume: 10
Electronic Location ID: e2308
Received 2024 Mar 25; Accepted 2024 Aug 15
Copyright: ©2024 Ullah Khan et al.
Copyright year: 2024
Copyright holder: Ullah Khan et al.
License: This is an open access article distributed under the terms of the Creative Commons Attribution License, which permits unrestricted use, distribution, reproduction and adaptation in any medium and for any purpose provided that it is properly attributed. For attribution, the original author(s), title, publication source (PeerJ Computer Science) and either DOI or URL of the article must be cited.
License URL: https://creativecommons.org/licenses/by/4.0/

Keywords: Internet of Things, Marine vehicle industry, MCDA

Funding: Institute of Information & communications Technology Planning & Evaluation (IITP) grant funded by the Korea government (MSIT) IITP-2017-0-00655 Lean UX core technology and platform for any digital artifacts UX evaluation This work was supported by the Institute of Information & communications Technology Planning & Evaluation (IITP) grant funded by the Korea government (MSIT) IITP-2017-0-00655, Lean UX core technology and platform for any digital artifacts UX evaluation. The funders had no role in study design, data collection and analysis, decision to publish, or preparation of the manuscript.

==============================
Rapid developments in the Internet of Things (IoT) have opened the door for game-changing applications in numerous sectors, especially the vehicle industry. There is a rising demand for efficient assessment and decision-making methodologies to pinpoint the most promising choices for the vehicle sector with the introduction of IoT-based maritime vehicles. To overcome this issue, the integrated multi-criteria decision-making analysis (MCDA) paradigm proposed in this research combines the additive ratio assessment (ARAS) and analytic hierarchy process (AHP) approaches to evaluate and choose IoT-based maritime vehicles based on their performance- and authenticity-related criteria in the vehicle sector. The selection issue is hierarchically organized, and the assessment criteria are prioritized using the AHP approach. There are seven performance and authentication related criteria are selected that might aid in the selection procedure. Using the AHP, we are assigned these criteria proportionate weights that reflect their respective significance and interrelationships. AHP, however, falls short of offering a thorough analysis of the alternatives that exist. To overcome these restrictions, this research presents the integration of AHP with the ARAS approach for the ranking of alternatives according to how well they perform against the set criteria. By using the ARAS technique, it is possible to get over the restrictions of AHP and achieve a more thorough assessment of maritime IoT-based vehicles. The efficiency of the framework is proven using empirical data and professional judgment. The findings show that the hybrid method successfully encapsulates the intricate relationships between the factors being evaluated and objectively appraises the potential of IoT-based maritime vehicles for the automotive sector. This study extends to the area by providing an organized and thorough method for assessing and choosing IoT-based maritime vehicles. Considering several factors and their mutual dependence, the hybrid AHP and ARAS technique gives decision-makers a powerful tool for evaluating the potential of IoT-based maritime vehicles in the automotive sector. Smart decisions on the deployment of IoT-based marine vehicles and maximizing the potential they present may be made by beneficiaries in the automotive sector using the study’s results.

Introduction

The maritime vehicles sector is going through a paradigm transition because of the Internet of Things (IoT) (Guo et al., 2023; Al-Atawi, Khan & Kim, 2022; Khan, Tarimer & Taekeun, 2022; Abbas et al., 2022; Al-Atawi, Khan & Kim, 2022; Gürüler et al., 2022), a network of connected devices that allows for smooth communication and data sharing. The incorporation of IoT technology into marine vehicles has given rise to IoT-based marine vehicles, which have significant promise for the automotive industry. These ships employ IoT capabilities to enhance the effectiveness, security, reliability, and sustainability of marine transportation. Sensors, actuators, and networking characteristics found in maritime IoT-based vehicles allow for real-time monitoring, data collecting, and analysis. They enable automated decision-making, autonomous operation, and sophisticated decision-making processes, revolutionizing several facets of the vehicle sector (Mohsan et al., 2023b). These vehicles have potential for use in maritime security, offshore tasks, industrial shipping, marine administrative tasks, and tracking environmental conditions. IoT-based maritime vehicles provide a number of potential benefits for the automotive sector. By offering current information on vessel effectiveness, energy usage, and repair needs, they first enable increased operational efficiency. This data-driven strategy improves the use of resources, route planning, and vessel optimization, resulting in cost savings and higher production using optical fiber (Ullah et al., 2018).

The automotive industry needs the vehicle ad hoc network, a cutting-edge technology since it can promote traffic effectiveness and security. Linked automobiles must provide significant data in an encrypted setting. Mohsan et al. (2023a) proposed a MARINE man-in-the-middle (MiTM) attack-resistant trusted architecture for networked automobiles that is effective at spotting malicious nodes running MiTM attacks and revoking their access rights. Each node executing the MARINE system first develops credibility for the sender by carrying out multivariate consistency tests on them. After the recipient has confirmed the sender’s validity, the received data are subsequently assessed in both direct and indirect ways. Several computations are conducted for an in-depth analysis of MARINE’s effectiveness and precision against three MiTM attacker types and the measured trust paradigm. In a network with 35% MiTM hackers, experiments demonstrate that MARINE outshines the most recent trust paradigm by increasing accuracy, recall, and F-score by 15%, 18%, and 17%, respectively.

For maritime IoT systems, Jung et al. (2023) have suggested an integrated low-Earth orbit (LEO) and unmanned aerial vehicle (UAV) computation solution. A significant amount of data gathered by ocean-connected devices may be promptly evaluated by two different edge servers that are deployed on LEO satellites and UAVs, respectively. By dynamically improving the bit management for connectivity and processing along with route guidance for the UAV under delay, energy financing, and working restrictions, the suggested system seeks to reduce the overall energy use of the battery-restricted UAV. The solutions are produced utilizing progressive convex estimation techniques for three distinct LEO satellite connectivity conditions—Always On, Always Off, and Intermediate Disconnected. By comparing the combined maximizing of bit distribution and UAV route scheduling with individual efficiency approaches that are designed for either the bit distribution or track of the UAV, the work shows that considerable reductions in energy usage are attainable in all LEO-accessible situations.

The major objective of the suggested review is to promote the deployment of onshore narrowband Internet of Things (NB-IoT) equipment for cargo monitoring on maritime cargo ships moving near the shore. The suggested study offers three connection choices, which are then evaluated. Two relay-aided devices employ direct sensor-to-onshore base station (BS) broadcast and direct sensor-to-UAV-mounted BS, respectively, as intermediary hubs. After utilizing probabilistic geometry to illustrate the procedure for connecting with the onshore installation, the proposed study uses system-level models to analyze and contrast the recommended solutions in terms of data loss and latency measures as well as device durations. The outcomes show that having an immediate access strategy performs the lowest. Due to the consistent distribution of the gearbox demands across time at the relay end and the elimination of congestion, relay-based systems significantly enhance system efficiency. Another benefit of UAV relaying is enhanced service, which raises the easily accessible BS density. The relaying procedure under investigation is robust to a variety of system variables (Kavuri et al., 2020).

To properly explain the effects of packet destruction and latency during transmission on the level of erroneous estimate, Lyu et al. (2020) present the Age of Information (AoI). The suggested investigation’s main focus is on the connection between the state erroneous estimation and the AoI of sensory stimuli. The research then looks at how estimated states and sensitive data transfer for maritime IoT devices may be integrated. The efficacy of the estimate is explicitly moderated by a mother ship (MS)-assisted collaborative distribution technique, which also explicitly reduces the effect of resource constraints and route loss. Then, the MS position, channel placement, and transmission power are simultaneously adjusted to reduce the average square error of state estimation. To do this, the breakdown procedure is used to formulate and solve a restricted elimination issue. The proposed technique is advantageous in terms of lowering estimate error and energy usage, according to testing findings.

Among the key innovations for investigating underwater assets is the intelligent control of autonomous marine vehicles (AMV). Cooperation amongst numerous AMVs can improve investigation efficacy in the deep sea with a challenging investigation setting by using a number of beneficial advantages. To solve this problem, Zhang et al. (2023) have proposed a strategy for cooperatively designing routes for diverse AMVs. For the surface route design of an autonomous surface vehicle (ASV) and the underwater route design of an autonomous underwater glider (AUG), this technique employs a hybrid metaheuristic technique. The suggested approach for underwater routing for challenging marine investigation is favorable, as evidenced by the enhanced convergence rate of the integrated metaheuristic system in experiments. Furthermore, marine vehicles powered by IoT also contribute to increased safety and reliability. They offer constant monitoring of crucial aspects including vessel status, weather, and navigational dangers by combining sensors and communication. Through early risk identification and prompt response made possible by continuous monitoring, accidents are decreased, and general maritime transportation safety is increased. The objectives of this research paper are as follows:

• To provide a detailed evaluation methodology that is intended to evaluate the alternatives of IoT-based maritime vehicles for the transportation industry.

• To determine and rate the assessment criteria that are important to the vehicle sector, considering factors like performance- and authenticity-related factors.

• To include the analytic hierarchy process (AHP) approach into the assessment process to create a hierarchical structure, define proportional weights to the assessment attributes, and capture the significance and linkages between them.

• To address the constraints of AHP and increase the assessment procedure by integrating the ARAS strategy, which utilizes a ratio-based strategy to grade the alternatives on the basis of their efficacy against the defined factors.

• To authenticate the precision by implementing the hybrid AHP and additive ratio assessment (ARAS) techniques into the assessment procedure to assess and choose IoT-based maritime vehicles for the transportation industry, utilizing real-world data and expert views.

• To give those in charge in the vehicle sector a powerful tool for assessing and choosing IoT-based marine vehicles based on their abilities, considering a variety of factors and their interconnections.

• To contribute to the body of research by highlighting how well the hybrid AHP and ARAS technique works for assessing and choosing innovative technologies, particularly IoT-based maritime vehicles, for the vehicle sector.

The typical sequence of sections followed in this paper includes: The first section introduces the research. ‘Literature Review’ explains related work in relation to this research. The third section describes the research methodology employed, detailing the steps involved in the hybrid AHP and ARAS method. In the fourth section, the findings of the evaluation are analyzed, interpreted, and discussed. Finally, the fifth section concludes the primary findings of the research and highlights its contributions.

Literature Review

The swift progress of artificial intelligence (AI) and the IoT may result in a significant number of self-sustaining and robotic surface vessels being a part of the quickly expanding marine transportation cyber-physical system (mTCPS). However, the rapid increase in the number of moveable vessels might result in subpar maritime effectiveness and security. Forecasts of movable vessel positions must be reliable and precise to improve intelligent traffic operations in IoT-enabled mTCPS. The outcome of the forecast can be utilized to enhance maritime tracking, identify unusual activity, and prevent crashes of ships. The enormous learning potential of deep learning serves as the driving force behind the study by Liu et al. (2021), which provides a smart system for informed vessel forecasting of trajectory relying on the well-known long short-term memory (LSTM). An effective trajectory forecast is made by combining the primary LSTM structure with a vessel traffic friction scenario model that was created employing real-time location data and the social power notion. A blended loss function is also developed to enhance the suggested structure for predicting trajectories under different laboratory circumstances.

The evaluations of real vessel trajectories demonstrate that the recommended method may enhance the forecasting of trajectories in terms of precision and stability. The Ocean of Things (OoT) architecture was created by Yang et al. (2019) to monitor the marine ecosystem leveraging IoT technologies. The three levels that comprise IoT are the data collection layer, the fog layer, and the cloud layer. Researchers utilize a mathematical gradient-based technique to analyze the initial collection data and finish the level of assurance of the device data collected. For multisensory knowledge combinations, a better D-S technique is created, decreasing storage space, and enhancing the integrity of data. The IoT system has various uses in the Intelligent Ocean because of the quick growth of computer technology. The speed of processing and broadcast integrity of sensory information was not possible because of the insufficient transmit power, long transmission separation, and complicated marine wireless settings, which would have an impact on the efficiency of environmental monitoring (Khan et al., 2023).

The research by Lyu, Chu & Lin (2021a) on collaborative multi-UAV broadcasting to increase overall throughput while bound by interruption probability, transmit strength, and channel bandwidth provides an idea for a solution to this challenge. Although the maritime ecosystem tracking program has a special requirement for the outage possibility, the suggested research specifically evaluates the future transmission dependability of the USV-OBS connection from the beginning. The research then mutually optimizes the USV-UAV interface and transmit power distribution to enhance the overall transmission efficiency while considering failures. The integer-programming challenge is then satisfactorily resolved using an intuitive technique. To take advantage of potential future advancement, Lyu, Chu & Lin (2021b) present the space-air-ground-sea incorporated network (SAGSIN) and connected tools, such as the shape-adaptive antenna and radar-communication convergence. The designed shape-adaptive aerial is constructed of lightweight components that enable it to alter its actual form in moments and generate a particular radiation beam by the operational demands. A thorough evaluation of the needs and practicality of radar wireless communication is done in addition to an exhaustive investigation and model validation for the aim of monitoring UAVs.

By merging several MTS components, Zhang et al. (2022) seek to create an IoT-based interactive computing system that integrates the modular design. Another crucial component of MTS is a distributed, regulated access strategy that cannot be altered or interfered with by outsiders. Blockchain technology is rapidly evolving and has become a crucial instrument for ensuring information safety because of its resilience to forging and manipulation. A cooperative computational system for IoT-based blockchain-based flow planning and oversight of marine transport is advocated by the study. The paper also suggests an unusual consensus method that uses reputation voting and Verifiable Random Function (VRF) to lower communication expenses in the blockchain consistency procedure. Muthuramalingam et al. (2019) have investigated and given a thorough account of the conception and construction of a prototypical IoT-based ITS infrastructure for an intelligent city situation, with a focus on the Indian subcontinent. The suggested investigation also addressed a range of system-related hardware and software elements. The fundamentals of big data analytics techniques such as the conjoint method, multiple regression technique, multiple discriminant investigation, and logistic regression have also been highlighted to advance the development of IoT-based ITS. Additionally, the case study has demonstrated some big data statistical findings and how they are used in digital infrastructure.

The study that was suggested offers a vehicle accident detection and categorization system (ADC) based on the IoT that combines onboard and internet-connected sensors from devices to both determine and notify the kind of accident. Knowing the extent of the individuals’ wounds and the extent of the car wreckage can help rescuers effectively manage the situation. To ascertain which ADC paradigm is most effective, the suggested study contrasts three machine learning algorithms that use naive Bayes, Gaussian mixture models, and decision tree approaches. Every prospective ADC system was trained and evaluated using five physical aspects of moving cars in an effort to decide which accident type—collision, rollover, falloff, and no accident—was most advantageous (Kumar, Acharya & Lohani, 2021). A portable YOLOX-s system with transferable knowledge proposed by Liu et al. (2022) initial recommendation for real-time boat size distinction. After that, more data is added to it to make the system more useful. An onboard smart edges-based augmented reality (AR)-based marine navigator is developed using multi-source sensor data, recognizable ships, and concurrent AIS signals. The AR system may apply both static and interactive data to the video-captured pictures by overlaying them with the acquired AIS transmissions. It can provide ASVs in MIoT systems with more details so they can have a head start on a navigational danger. In comparison to conventional single-sensor-based navigational devices, the data fusion architecture looks to have many useful uses and produce higher-quality and solid outcomes. The multiple testing indicated that the model works better in several distinct navigating scenarios.

Constantinou et al. (2021) developed a framework for simulating aspects of the Internet of Things (IoT) that connect data gathering effectiveness to resource usage. The simulation was used to maximize the utilization of limited resources while improving the efficiency of data collecting. The effectiveness of the suggested approach for classifying underwater noises was shown by utilizing simulation, laboratory, and field research. IoT and big data technologies are expanding swiftly. A newly emerging IoT sub-paradigm called Industrial IoT (IIoT) is primarily focused on commercial uses that require safety. IoT helps organizations better understand and manage their assets, operations, and surroundings; big data insights and IIoT co-evolve as a double “helix”; and embracing big data insights has been proven to improve organizations’ profitability and efficiency. Wang et al. (2015) evaluated the possibilities and obstacles for the maritime cluster in this big data and IoT age before offering a unique design for offshore support vessels (OSVs) that uses a hybrid CPU/GPU/FPGA1 high-performance computation architecture. The model presented by the study might assist marine firms boost their production and efficiency when put into use, enabling the entire cluster to keep its position as a pioneer in the international maritime sector.

Numerous negative consequences on the atmosphere and humans have been caused by the steady growth of plastic in maritime habitats. Inadequate disposal procedures, growth in the usage of plastics, and the durability of plastics are the causes of this buildup. In these situations, sending data across a cloud platform for later collection and analysis using certain statistical techniques is the norm. In demanding situations with inconsistent internet access, such as aquatic environments, drone surveillance networks, and satellite communication systems, the fundamental data transmission process becomes difficult. Montella, Ruggieri & Kosta (2018) developed a revolutionary IoT data transmission paradigm for compassionate cloud-based apps. Because it is (1) elastic, using quick associations, (2) lightweight, ideal for IoT devices, and (3) secure, the architecture’s interface is an excellent option for services that manage confidential information. To demonstrate the system’s advantages in terms of efficiency, sturdiness, and security, the proposed research examines it in an actual setup with maritime navigation software.

To improve the system’s sum rate, the suggested study creates an integrated power distribution difficulty. In contrast to previous research, the proposed strategy exclusively employs large-scale channel state information (CSI) for system complexity minimization. The geographical data given by marine IoT devices are used to construct a whole CSI. The issue cannot be solved due to nonconvex restrictions. The suggested study uses sequential convex estimation, an auxiliary function approach, and max–min efficiency to overcome these challenges. In response, a progressive energy distribution method has been suggested. Analyses indicate a considerable improvement in coverage. This demonstrates the capability of blended NOMA satellite-UAV-terrestrial infrastructures to provide marine streaming services (Fang et al., 2022). The proposed research offers a thorough examination of the most significant maritime communication techniques as well as the most recent advancements in a range of marine technology. The first part of the proposed study covers radio frequency (RF) and optical band marine messaging systems.

Alqurashi et al. (2022) then discuss network designs for RF and optical bands, transmission and coding methods, protection and ability, and radio scheduling in marine operations. The study endeavor then discusses a few recent advancements in maritime systems, like the Internet of Ships and the ship-to-underwater Internet of Things. The proposed study also suggests prospective paths for studies on maritime interaction such as expanding internet access to deep-sea environments, utilizing noticeable and terahertz signals for onboard uses, and using information-driven simulations for radio and optical marine transmission. The survey also includes several fascinating open-ended queries. Essential safety measures are put in place to lessen the negative consequences of electrical machinery and gadgets with poor electrical performance. The dependability and security of the power source have a considerable influence on the personnel and security of any ship, and the certified Contextual Data Prediction Technology (CDPT) is used to protect and monitor ship propulsion. The sensing device has been utilized to examine the motor functioning using a CDPT controller and then transmit the information to the consumer using IoT to simulate an ordinary propulsion drive on a completely electrical cruise ship. The tool was then used to construct the entire framework. Thus, from the standpoint of the vessel’s system, the updated model successfully expanded, and the computational findings showed that it was a very accurate model (Su et al., 2021).

Understanding and assessing social and environmental dynamics while employing restrictive data-gathering techniques is difficult in the context of maritime monitoring. Agile distributed measurement and processing that supports scalability, variety in data collection, and responsive deployment of essential assets was required due to the nature of the issue and the paucity of assets, including energy, interpreting, and connectivity. Zadorozhnyi et al. (2021) method allows them to follow several military warships by using data from a variety of sensors. A fusion core with highly computational powers and an ecosystem of flexible marine IoTs with little computing power work together to deliver precise surveillance. The core processing decisions made by the IoTs can be changed to sporadically generate useful metrics. The diverse information fusion center manages the challenging task of determining vehicle state. The results show that the suggested system may drastically cut working and bandwidth requirements without degrading monitoring efficiency by altering its adaptive collaborative computing features. The study proposed by Moghimi & Mohanna (2023) contains a substantial number of low-resolution video frames of several aquatic organisms and tackles prediction-relevant issues. Submarine IoT devices employ an updated convolutional neural network (CNN) design to allow MTS through the skillful manipulation of retrieved visual computing components. The recommended method extracts learned submerged components from video frames using an updated residual neural network (mResNet). The mResNet method incorporates the dilation characteristic to gather as much data from the video clips as it can. The SMARTSEA methodology developed by Katranas et al. (2020) intends to train workers and pupils for careers in the emerging digital maritime and surveying sectors. Another goal is to promote multifaceted abilities like a stronger sense of innovation and creativity. Additionally, a cutting-edge method of evaluating the idea throughout the entirety of Europe is shown.

Problem statement

Marine vehicles powered by IoT also have prospects for sustainable development. These vehicles provide optimized energy use, fewer emissions, and effective resource utilization through data-driven understandings. They support the adoption of environmentally beneficial practices such as route optimization, pollution monitoring, and preventative care, which is in line with the growing need for environmentally friendly and sustainable practices in the automotive sector. Successful assessment and selection techniques are essential for maximizing the capability of IoT-based maritime vehicles. The rapid development of the IoT has led to the introduction of IoT-enabled marine vehicles, which have significant effectiveness in the auto industry. However, reliable evaluation and decision-making processes are required to identify and rank the most feasible IoT-powered marine vehicles. This study addresses this difficulty by introducing a hybrid technique combining the AHP and ARAS to analyze and choose marine IoT-based boats according to their integrity and performance-centric criteria. The objective is to serve decision-makers in the vehicle sector with an accurate resource for the assessment and determination of maritime vehicles that are IoT-based while considering critical performance and authenticity related factors and their mutual dependence.

Contribution based on existing literature

This study attempts to close a substantial gap in the literature by integrating the ARAS and AHP to analyze secure authentication methods in IoT applications within the marine sector. Although ARAS and AHP are well-known methodologies, our contribution is the way we use and integrate them in a particular yet unexplored area: the maritime sector. As far as we have learned from the existing studies, there aren’t any appropriate selection procedures specifically designed for the challenge of choosing secure and authentic IoT solutions in the marine industry at this moment. An in-depth review of several topics, such as IoT strategy and authentication system preference, reveals that there aren’t many evaluation mechanisms made especially for this industry. However, the research does more than simply encourage the utilization of tried-and-true methods. Consequently, the proposed approach is thoroughly analyzed and contrasted in this paper with different appraisal mechanisms used in various industries, such as entropy, analytic network process (ANP), Technique for Order of Preference by Similarity to Ideal Solution (TOPSIS), and Decision-Making Trial and Evaluation Laboratory (DEMATEL). Prioritization should be given to describing the relative advantages that our approach provides and identifying topics for enhancement that are unique to the marine industry. As a result, the proposed study improves upon the already adopted assessment methodologies and offers a feature-focused analysis relevant to IoT-based marine systems. Thus, our research’s comprehensive analysis seeks to close existing gaps and provide a solid theoretical and numerical framework for both scholars and practitioners. The published studies cited from numerous libraries have been depicted in Fig. 1.

Figure 1 Related studies from different libraries.

Methodology

The rapid usage of IoT brings numerous advancements to every sector including education, medical, and marine field. Maritime vehicles are one of these fields, which widely adopted IoT technology for the advancement of their services and automation. There is very little work found on the selection and evaluation of IoT-based marine vehicles for the potential of the vehicles industry using hybrid AHP and ARAS techniques. This study develops a hybrid approach for the comprehensive evaluation and appropriate selection of IoT-based marine vehicles that will be very helpful for the potential of the vehicles industry in the future. Several essential features related to decision-making have been identified and selected to weigh them and determine their relative importance for alternative ranking. The criteria have been weighted using AHP, whereas the alternatives were ranked using the ARAS method. All the alternatives have been ranked and thus a significant one is determined. The purpose of the article is to enhance academics’ comprehension of the potential of multi-criteria decision making (MCDM) methods in the field of energy. MCDM techniques were employed to perform further mathematical calculations and look for patterns over several years. This broad inquiry also covers the most current changes in the energy sector (Kaya, Çolak & Terzi, 2018). The major goal of the proposed study (Goswami & Behera, 2021b) is to underline the crucial nature of identifying the optimum goods from a list of seven options based on six different significant factors. Multi-criteria decision-making analysis (MCDA) methods were used to analyze these requirements. The most beneficial option has been determined, and the ARAS approach was used to recommend an appropriate sorting sequence for the potential outcomes. In this instance, the entropy approach was used to calculate the weights of the criterion. According to the gauge, the least favored alternatives include cast iron, carburized steel, and hardened alloy steel. Cast alloy steel was the finest alternative. Improved potential ratings were additionally stated, and a new study was conducted using the outcomes of the prior studies. Among the main causes of global warming is the cutting agent utilized during metalworking. Therefore, it is essential that you carefully choose a green fluid to drastically decrease harmful emissions. Goswami & Behera (2021a) have chosen the optimal green cutting fluid out of three options using the COmplex PRoportional ASsessment (COPRAS) and ARAS approaches. The numerical values for the criterion were created using the analytic hierarchy technique. Cost, environmental effects, and quality were a few factors that should be considered before making a decision. Quality must be increased while expenses and environmental effects must be decreased. The ideal and poorest options, according to COPRAS and ARAS, were Syntilo 9930c and conventional cutting fluid. The discoveries of the earlier study’s investigation were precisely replicated by the alternative ranks, which were also evaluated.

The AHP and ARAS represent two fundamental tools utilized in the decision-making processes within IoT-enabled maritime vessels. ARAS is recognized for its straightforwardness, rendering it appropriate for swift decision-making tasks and the assessment of variables such as safety, efficacy, and cost-efficiency. Its capacity to adapt to various forms of data facilitates a comprehensive examination of both subjective and objective components. Conversely, AHP employs a hierarchical structure to arrange intricate decision-making processes into more manageable levels. This methodology proves particularly beneficial in the maritime sector, where decisions encompass a multitude of considerations across different levels. The pairwise comparison method further enriches decision-making by methodically comparing criteria and alternatives, prioritizing elements according to their importance. AHP also integrates consistency checks to guarantee the reliability of the process, diminishing the chances of biases or inconsistencies in intricate decision contexts. In essence, the amalgamation of ARAS and AHP establishes a resilient decision-making framework for IoT-based maritime vehicles, elevating the quality of decisions, fostering transparency, and encouraging engagement from stakeholders.

Features engineering

The formulation of comparative parameters for selecting dependable IoT techniques to address security and autonomous concerns in marine industrial contexts forms the basis of the recommended design. characteristics focusing on identity and access control concerns are defined to guarantee that only permitted gadgets may access the IoT network. The fact that protection measures encompass every aspect of IoT, and autonomous technologies is the primary rationale in favor of adopting them. The attributes of IoT technologies are constructed from a variety of sources. A thorough and precise investigation of the literature is performed to uncover and gather the necessary attributes for the comparison of IoT devices integrated into marine vehicles. These attributes are quite ubiquitous since numerous different authentication systems employ them. After the initial identification of sixty-seven attributes from diverse sources, there are now just seven attributes left, allowing for examination and selection among many other options. In Fig. 2, all the attributes from the literature study are described.

Figure 2 List of all characteristics.

Identifying and selecting features

The suggested evaluation approach’s initial level lays the greatest emphasis on selecting qualities and alternatives. The information from an IoT expertise panel is contrasted with seven viable solutions, including V1-V7. The efficacy of maritime vehicle technologies will be assessed in an IoT environment. Only the option that earned the highest overall rating out of all feasible options considering the established criteria would be regarded as the best choice in terms of their chosen criteria. This approach will work or be regarded as authentic depending on the attributes of each of the seven features-based criteria. These seven criteria consist of safety features (F1), navigation accuracy (F2), environmental impact (F3), transmission throughput (F4), authenticity (F5), energy efficiency (F6), and operational cost (F7), as illustrated in Fig. 3. The effectiveness of the criterion depends on how all of them interact with one another.

Figure 3 List of selected features.

Evaluation framework based on multicriteria approaches

The AHP and ARAS methodologies operate as assessment instruments and selection options to choose the best target solution for maritime IoT-based vehicles. The fusion of ARAS and AHP within marine optimization presents notable advantages attributable to their synergistic interrelation. ARAS is characterized by its simplicity and adaptability, facilitating prompt evaluations in decision-making processes, whereas AHP excels in organizing intricate decision variables and prioritizing alternatives based on their significance. This methodology proves particularly valuable in the evaluation of multifaceted elements such as operational efficiency, safety, environmental consequences, and cost-effectiveness. The fusion of ARAS and AHP amplifies risk management by streamlining comprehensive risk assessments and prioritizing strategies for risk mitigation. AHP’s validation procedures ensure coherence and diminish biases in decision-making processes, which are essential for the evaluation and mitigation of risks inherent in maritime operations. Furthermore, this integration fosters engagement with stakeholders and sustains transparency in decision-making protocols. Through the incorporation of ARAS’s simplicity and AHP’s structural framework, decision-makers can arrive at well-informed, data-driven decisions that account for diverse variables and perspectives of stakeholders. This integration enhances the quality of decisions across various facets of marine optimization, thereby aligning with objectives such as efficiency, safety, sustainability, and stakeholder contentment. In addition, the envisioned appraisal framework’s three primary stages are as follows: The initial step is integrating the attributes into the testing scheme. In the second computing stage, numerical computations are used to determine the alternatives’ intrinsic attributes. When all empirical evidence comes in, the most efficient choice is picked as the IoT solution using the ARAS method based on the scores given for the criterion. Below is a complete mathematical calculation and processes involved in the AHP and ARAS techniques.

AHP approach

This MCDM technique works well for organizing and analyzing complicated choices. AHP offers a practical method for handling several issues pertaining to various areas while making precise and practical selections. Saaty initially introduced this method in 1980, and by utilizing the presented Saaty scale (Fang et al., 2022), we can choose wisely from the numerous options that are accessible based on a number of distinct factors. The current study applies the AHP approach to the chosen parameters to determine how much every attribute is weighted. Using these weights, ARAS ranks several computer-based health monitoring applications to improve the performance of the healthcare sector. Figure 4 illustrates all the phases required to complete the procedure to determine the weights of the chosen attributes using AHP.

Figure 4 AHP steps.

Step 1. AHP hierarchical model

This presents the issue as an ordered framework with three levels. The goal is displayed at level 1, and the criteria are displayed at level 2. The final level is where the choices are addressed, as seen in Fig. 5.

Figure 5 AHP hierarchical diagram.

Step 2. Pair-wise comparison matrix representation of features

Diagrammatic notation cannot be utilized for computational analysis but is best suited for visual inspection. Like this, when a framework gets bigger it is more difficult to comprehend the accompanying graphic visually. Therefore, it is essential to provide a description that is easy to comprehend so that computers can successfully store, retrieve, and analyze data. Continuing with the Saaty scale, a matrix is created by assigning particular scores to each factor per the expert’s needs. Finally, a pair-wise assessment matrix of size 7 × 7 is created and assembled for the present situation. The comparative weights of each criterion obtained from the IoT experts using a scale are as displayed below in the matrix. CriteriaF1F2F3F4F5F6F7F11.003.002.002.000.333.003.00F20.331.002.000.330.503.002.00F30.500.501.000.500.330.502.00F40.503.002.001.002.003.003.00F53.002.003.000.501.003.002.00F60.330.332.000.330.331.000.50F70.330.500.500.330.502.001.00.

Step 3. Normalizing comparison matrix and criteria weights

The matrix of pairwise comparison is normalized for each criterion using Eq. (1). (1) Nij=XijSumofeverycolumn.

The comparative matrix is normalized to eliminate subjectivity. Then, the relative importance of every criterion is figured out by averaging the normalized comparative matrix for every row in accordance with Eq. (2). (2) CrieriaWeights=∑Nijn.

The determined normalized matrix and criterion weights are provided in the below matrix.

CriteriaF1F2F3F4F5F6F7F10.170.290.160.400.070.190.22F20.060.100.160.070.100.190.15F30.080.050.080.100.070.030.15F40.080.290.160.200.400.190.22F50.500.190.240.100.200.190.15F60.060.030.160.070.070.060.04F70.060.050.040.070.100.130.07Sum1111111 CriteriaWeights0.210.120.080.220.230.070.07.

Step 4. Multiplication of pairwise comparison matrix and criterion weights

The pair-wise assessment matrix and the derived criterion weights are multiplied to assess the precision of the estimated value. The required process is as illustrated below in matrix form.

CriteriaF1F2F3F4F5F6F7F11.003.002.002.000.333.003.00F20.331.002.000.330.503.002.00F30.500.501.000.500.330.502.00F40.503.002.001.002.003.003.00F53.002.003.000.501.003.002.00F60.330.332.000.330.331.000.50F70.330.500.500.330.502.001.00×CriteriaWeights0.210.120.080.220.230.070.07.

Step 5. Results of the above product matrix and calculation of the weighted sum value

The required outcomes figured out by the above multiplication matrix are as provided. Furthermore, the total of every row in the generated matrix is computed to get the weighted sum score. The matrix having the multiplied values and the calculated weighted sum scores from it are as illustrated below. CriteriaF1F2F3F4F5F6F7F10.210.350.160.440.080.210.22F20.070.120.160.070.110.210.15F30.110.060.080.110.080.030.15F40.110.350.160.220.450.210.22F50.640.230.240.110.230.210.15F60.070.040.160.070.080.070.04F70.070.060.040.070.110.140.07WeightedSumValue1.670.890.611.721.810.520.57.

Step 6. Identification of the ratio of weighted sum value and attributes weights

To calculate the ratio of the weighted sum score and criteria weights, the weighted sum scores were divided by the criterion weights. In Table 1, the estimated ratio between the weighted aggregate score and the criterion weights is as displayed.

Table 1 Ratio Computation of weighted sum score and attributes weights.

Criteria	Weighted sum score	Criteria weights	Weighted sum score/Criteria weights	
Safety features (F1)	1.67	0.21	7.80	
Navigation accuracy (F2)	0.89	0.12	7.58	
Environmental impact (F3)	0.61	0.08	7.67	
Transmission throughput (F4)	1.72	0.22	7.76	
Authenticity (F5)	1.81	0.23	8.03	
Energy efficiency (F6)	0.52	0.07	7.61	
Operational cost (F7)	0.57	0.07	7.73	

Step 7. Calculation of λmax, CI, and CR

(3) λmax=7.80+7.58+7.67+7.76+8.03+7.61+7.737.

To calculate the value of the Consistency Index (C.I.), we apply the following Eq. (4) (4) C.I=λmax−nn−1,

while n stands for the number of attributes/criteria.

C.I=7.74−77−1

C.I=0.12.

To figure out the Consistency Ratio (C.R), we apply Eq. (5) (5) C.R=C.IR.I.

While RI stands for the Random Index, the value of RI is 1.32 at the seven positions. CR=0.121.32.

CR = 0.09, Hence the value of CR is acceptable because it is less than 0.1.

Figure 6 visually stands for the criterion weights that were identified by the AHP method.

Figure 6 Criteria weights based on AHP.

ARAS approach

The Additive Ratio Assessment (ARAS) technique is used by many industrial sectors, to help with making decisions. The ARAS technique chooses “the finest” choice by streamlining complicated decision-making situations. The relative signal could draw attention to distinctions between the suggested course of action and the available choices. This is carried out by taking different measuring units’ effects out of the equation (Alqurashi et al., 2022). The incorporation of ARAS is directed towards addressing the constraints encountered in the utilization of AHP for decision-making purposes. Although AHP demonstrates proficiency in structuring hierarchies and conducting pairwise evaluations, it encounters challenges when handling extensive decision matrices and aiming for simplicity. ARAS, recognized for its flexibility, is deemed appropriate for prompt assessments and the management of diverse criteria without unnecessarily complicating the procedure. The suggested methodology strives to leverage the strengths of both approaches, simplifying the process and presenting a clear-cut evaluation framework. ARAS’s additive characteristic fosters transparency and the participation of stakeholders, thereby enriching decision-making quality, effectiveness, and alignment with the objectives of maritime operations. The ARAS data analysis technique includes weights and criterion additions during the computation phase to aid the IoT experts and industrialists in making an optimal decision practical. The ARAS approach is used as an analytical instrument and decision-making process to choose the best intended alternative for a secure and effective IoT-based marine transportation system. IoT experts were enlisted, and as stated before, AHP was used to classify performance and authentication-related features. The highest-ranking alternative is then selected as the target authentication solution based on the specified attribute values. An illustration of the steps involved in this method used to evaluate and rank the intended strategy for IoT-based marine transportation systems is as shown in Fig. 7. This section provides information on the steps and details for ARAS.

Step 1. Decision matrix representation of features and alternatives

The graphic depiction is most suited for visual screening; nevertheless, computational analysis cannot be done with it. The accompanying graph also grows complex for big systems, which therefore affects how easily it can be understood visually. Therefore, it is crucial to provide a model that is simple to understand so that computers can efficiently gather, access, and analyze data. The entire set of chosen attributes is beneficial in this process. The performance attributes and data gathered from IoT and marine vehicle specialists are provided in a decision matrix with a dimension of 7 by 7. Seven choices are chosen for selection based on the defined attributes in this matrix, which includes seven performance factors listed in columns. The optimum value (OV) in each column has also been found. The viewpoints gathered from the IoT specialists are separated into seven choices as shown in the decision matrix below based on the weights provided to each attribute. CriteriaAlternativesF1F2F3F4F5F6F7V12.005.003.007.006.004.008.00V24.007.002.005.003.006.002.00V36.003.005.002.004.007.009.00V43.002.006.004.008.005.006.00V55.003.004.003.009.002.007.00V68.006.002.007.005.003.004.00V77.004.006.008.002.005.003.00OV8768979

Step 2. Normalizing a decision matrix

The selection matrix has been normalized for both beneficial and non-beneficial attributes. Greater scores are preferred for the selection matrix’s attributes since all of them are beneficial. The formulas listed below are used to figure out factors that are both beneficial and not.

Figure 7 ARAS steps.

For beneficial criteria, (6) X¯ij=Xij ∑i=0mXij

For non-beneficial criteria,

(7) Xij=1Xij;X¯ij=Xij∑i=0mXij.

The selection matrix has been normalized using Eq. (6) to eliminate variability because all the chosen attributes are beneficial in nature. Then, the criterion weights figured out by AHP will be multiplied with it to get the weighted normalized matrix in the next step. The normalized selection matrix’s specifics are mentioned below in the matrix form.

CriteriaAlternativesF1F2F3F4F5F6F7V10.050.140.090.160.130.100.17V20.090.190.060.110.070.150.04V30.140.080.150.050.090.180.19V40.070.050.180.090.170.130.13V50.120.080.120.070.200.050.15V60.190.160.060.160.110.080.08V70.160.110.180.180.040.130.06OV0.190.190.180.180.200.180.19 * CriteriaWeights0.210.120.080.220.230.070.07

Step 3. Weighted normalized decision matrix

The factor values acquired through the application of AHP are multiplied with every factor of the normalized selection matrix using Eq. (8) to derive the weighted normalized selection matrix. The results obtained from this calculation are provided in matrix form as given below. (8) X ˆij=X¯ij∗WjCriteriaAlternativesF1F2F3F4F5F6F7V10.0100.0160.0070.0350.0290.0070.012V20.0200.0220.0050.0250.0150.0110.003V30.0300.0100.0120.0100.0200.0120.014V40.0150.0060.0140.0200.0390.0090.009V50.0250.0100.0090.0150.0440.0040.011V60.0400.0190.0050.0350.0240.0050.006V70.0350.0130.0140.0400.0100.0090.005OV0.0400.0220.0140.0400.0440.0120.014

Step 4. Pi-Optimality function for ith alternative and calculation of utility degree

Equation (9) has been used to get the values of the optimality function (Pi) for every option as the aggregate of each row of the weighted normalized matrix was determined. (9) Pi= ∑J=1nX ˆij;i=0¯,m.

For the determination of the utility degree (Ki)

The comparison of the alternative with Eq. (10), yields the Ki for different alternatives. (10) Ki=PiP0

while the value of Po is 0.187 as well.

The results obtained by these equations are as shown in Table 2.

Step 5. Finalize the ranking of alternatives

The options are sorted in an ascending sequence of Ki. The most desirable option is the one with an elevated Ki value, whereas the most undesirable option is the one with the smallest Ki score. The calculated ranking of each alternative based on the ARAS method is as shown in Fig. 8.

Results and Discussion

The advent of IoT-based maritime vehicles is causing an evolutionary shift in the automotive sector. By using IoT technology, these vessels improve maritime transportation’s effectiveness, security, reliability, and long-term viability They have sensors, communication, and data analytics features, which make it possible to track in real-time, plan for maintenance, route more efficiently, and make wise decisions. As IoT-based marine vehicles become more prevalent, there is an increasing need for thorough review and decision-making techniques to determine the vehicle sector’s most viable options. To overcome this issue, the study thoroughly assesses the IoT-based marine vehicles to select a suitable alternative for the potential of the marine transportation sector using hybrid AHP and ARAS methods. Initially, the method of evaluating IoT-based marine vehicles entails selecting and comparing the most important assessment criteria. These factors consist of performance- and authentication-relevant factors, to figure out their relative weights. The AHP technique, which offers a formal framework for assessing options based on a variety of factors, is then used. To perform this, a hierarchical structure was created, pairwise comparisons were made to learn the relative relevance of the factors, and weights were assigned to each criterion. AHP helps to make knowledgeable assessments by measuring the factors to consider and their mutual dependence. By resolving pairwise comparison deficiencies, ARAS enhances AHP. It uses a ratio-based method to order alternatives according to how well they perform in comparison to predetermined criteria. ARAS provides a thorough evaluation of the options, considering both beneficial and detrimental features, using additive value functions and preference ratios.

Table 2 Optimality function, utility degree, and ranking of alternatives.

Alternatives	Pi	Ki	Ranking	
	0.187	1		
V1	0.117	0.626	4	
V2	0.100	0.538	7	
V3	0.107	0.573	6	
V4	0.113	0.604	5	
V5	0.117	0.628	3	
V6	0.135	0.722	1	
V7	0.125	0.671	2	

Figure 8 Ranking of alternatives based on ARAS.

The hybrid method used in this work, which combines the AHP and ARAS approaches, offers various benefits in the accurate evaluation of IoT-based maritime vehicles. The selection and ranking of pertinent assessment criteria are a part of the suggested evaluation procedure. These criteria, which include factors relating to performance and authentication, are essential for figuring out the viability of IoT-based maritime vehicles. AHP offers a hierarchical and systematic framework for grouping assessment criteria, identifying their connections, and determining relative weights. However, by using a ratio-based methodology, ARAS overcomes the drawbacks of AHP’s pairwise comparisons and offers a more thorough analysis of the available options by ranking the available alternative in ascending order. The entire findings determined by the AHP show that the authenticity factors receive the highest relative weight of 0.23, following the transmission throughput of 0.22, safety features of 0.21, navigation accuracy of 0.12, environmental impact of 0.08, and the energy efficiency and operational cost have the same weight of 0.07 as shown in Fig. 9.

Figure 9 Weights of criteria based on AHP.

Based on the determined weights of criteria using AHP, alternatives have been ranked using the ARAS method. The assessment of the alternatives indicates that the alternative V6 secures the highest optimality function score of 0.135 and utility degree of 0.722 and becomes the best alternative among others, followed by the V7 placed 2nd with values of 0.125 and 0.671, the V5 placed at 3rd with values of 0.117 and 0.628, the V1 placed 4th with optimality function of 0.117 and utility degree of 0.626, the V4 placed 5th with scores of 0.113 and 0.604, the V3 placed 6th with scores of 0.107 and 0.573, and the V2 placed at the end of the list with optimality function of 0.100 and utility degree of 0.538 and becomes the worst alternative as shown in Fig. 10. In general, the overall outcomes show that the V6 alternative was the best-performing alternative, among others. These results indicated that the V6 performed better in the specified set of attributes and has optimal one. Thus, all the chosen alternatives against the set of criteria have been ranked and the best-performing one is found.

Figure 10 Alternatives ranking determined by ARAS.

The analysis outlines the practicality of implementing a decision-making framework within the realm of maritime vehicles, focusing specifically on the automotive sector. It posits that the integration of AHP and ARAS offers utility across diverse industries such as aviation, transportation logistics, renewable energy, and manufacturing. This holistic approach has the capacity to optimize operational effectiveness, resource distribution, risk management, and strategic decision formulation. In the aviation domain, the ARAS-AHP methodology can be employed to prioritize maintenance procedures, optimize flight paths, and oversee aircraft fleets. When it comes to transportation planning, this strategy can streamline supply chain operations, aid in selecting appropriate modes of transportation, and optimize warehouse management. The renewable energy field can derive advantages from the ARAS-AHP approach in tasks such as site selection, evaluation of energy production technologies, and assessment of environmental impacts. Similarly, the manufacturing industry can utilize this approach to enhance production processes, choose suitable suppliers, and develop effective product strategies. The research underscores the flexibility and scalability of this methodology across various sectors, underscoring its significance and transferability among decision-makers and stakeholders.

Decision-makers are given a solid foundation for evaluating the advantages and disadvantages of these vehicles by the hybrid AHP and ARAS techniques used to evaluate the potential benefits of IoT-based maritime vehicles for marine transportation. The outcomes of the assessment show the benefits and potencies of IoT-based maritime vehicles. In addition to greater energy efficiency and route planning, these vehicles also displayed real-time data gathering for more informed choices. Additionally, they demonstrated improved levels of safety thanks to sophisticated sensor systems, continuous monitoring of crucial variables, and adaptive risk reduction techniques. The analysis revealed that IoT-based marine vehicles provide greater dependability and cost-effectiveness thanks to strengths for preventative upkeep and optimized resource utilization. Stakeholders may consider a variety of factors and how they interact when determining whether to deploy and integrate IoT-based marine vehicles. This assessment technique serves the marine vehicles industry by enhancing performance, reliability, effectiveness, and long-term viability. For the marine vehicles sector, the assessment of IoT-based maritime vehicles utilizing a hybrid AHP and ARAS methodology has important real-world consequences. The evaluation results may be used by those in charge to guide their strategies for deploying new technologies, their strategic goals, and their investment decisions. Stakeholders may choose which IoT-based marine vehicles will best suit their needs thanks to the insights acquired during the screening procedure. It is necessary to consider the constraints put forward by the testing technique and the facts at hand. Establishing subjective weight choices and finding trustworthy data sources might be difficult. Future studies might concentrate on improving the hybrid AHP and ARAS approach, looking into new assessment criteria, and assessing the long-term effects of using IoT-based maritime vehicles in the marine transportation sector. Also, the inclusion of case studies or simulations can further strengthen its practicality and promote its adoption across different sectors. The research results offer insightful information to those involved in the marine vehicles sector, aiding in decision-making, and directing the embrace and incorporation of IoT-based maritime vehicles in the future.

Conclusions

This research used the AHP and ARAS methods to assess the potential of IoT-based maritime vehicles in the marine vehicles sector. The study was effective in achieving its goals, and it has significantly advanced the discipline. The assessment methodology used for this study offered a methodical and thorough way to evaluate the possibilities of maritime IoT-based vehicles. The AHP technique was implemented into the context to generate a hierarchy of levels and prioritize assessment criteria, including factors linked to performance and authentication. This research appropriately reflected the significance and interconnections of these factors by putting relative weights on them through the AHP strategy. The ARAS approach is then added to the hybrid architecture to improve the assessment procedure. To rate the alternatives according to how well they performed in comparison to the predetermined criteria, ARAS used a ratio-based method. This incorporation of AHP-based ARAS gave a more thorough assessment of the options and solved the shortcomings of AHP’s pairwise comparisons. Furthermore, the combination of AHP and ARAS techniques was used to assess IoT-based maritime vehicles in the marine transportation sector, proving its efficacy. The outcomes demonstrated the strategy’s capacity to capture complicated relationships among the assessment criteria using real-world data and expert views. Because of their potential in the maritime vehicle manufacturing sector, the screening procedure effectively pinpointed the most attractive alternatives. The results of this study have important ramifications for those who make decisions in the marine vehicle sector. By considering many factors and their mutual dependence, the assessment framework offers a reliable instrument for evaluating the potential of IoT-based maritime vehicles. It enables stakeholders to embrace the potential advantages provided by IoT technology by empowering them to make knowledgeable decisions about the acceptance and incorporation of these revolutionary maritime vehicles. It is concluded by the research findings that this study advances the subject by providing a structured and precise methodology for assessing IoT-based maritime vehicles for their potential in the marine transportation sector. The hybrid AHP and ARAS technique gives decision-makers a solid foundation for making decisions, allowing them to evaluate and choose the best alternatives. The study’s findings can help industry participants make better decisions so they can take advantage of the prospects provided by IoT-based marine vehicles and progress the marine transportation sector. It is concluded that the study lays the groundwork for future analysis and selection of IoT-based maritime vehicles for the marine vehicle sector.

There are numerous future directions and limitations that need to be acknowledged even though the research provides significant knowledge into the analysis and selection of IoT-based marine vehicles in the vehicle industry. Future research might first focus on incorporating real-time data and predictive analytics into the evaluation process to improve the precision and rapidity of decision-making. A more in-depth assessment paradigm might be created by including additional factors like affordability, regulatory compliance, and user acceptability in addition to the performance- and authenticity-related factors that the research focuses on. Furthermore, the suggested hybrid AHP and ARAS technique is based on expert opinion and scientific evidence, and it would be more applicable if real-world scenarios and case studies were used to validate it. By using the hybrid assessment framework in diverse situations and proving its efficacy in a range of circumstances, more studies can elaborate on the course of this research. The assessment procedure can also be improved further by looking at integrating new decision-making techniques or assessment criteria. By considering these future paths and constraints, scholars and policymakers may persist in accelerating the enhancement of the marine vehicle sector and promoting sustainable growth within the vehicle manufacturing industry.

Supplemental Information

Supplemental Information 1 Calculations performed using Excel formulas for AHP and ARAS

Additional Information and Declarations

Competing Interests

Author Contributions

Data Availability

The authors declare there are no competing interests.

Habib Ullah Khan conceived and designed the experiments, performed the experiments, analyzed the data, prepared figures and/or tables, authored or reviewed drafts of the article, supervison, and approved the final draft.

Muhammad Abbas conceived and designed the experiments, performed the experiments, analyzed the data, performed the computation work, prepared figures and/or tables, authored or reviewed drafts of the article, and approved the final draft.

Shah Nazir conceived and designed the experiments, performed the experiments, analyzed the data, performed the computation work, prepared figures and/or tables, authored or reviewed drafts of the article, and approved the final draft.

Faheem Khan conceived and designed the experiments, performed the experiments, analyzed the data, prepared figures and/or tables, authored or reviewed drafts of the article, and approved the final draft.

Jamil Hussain analyzed the data, performed the computation work, authored or reviewed drafts of the article, and approved the final draft.

The following information was supplied regarding data availability:

Attached in the Supplementary File as code.

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
