# Peer review of "Optimizing marine vehicles industry: a hybrid analytical hierarchy process and additive ratio assessment approach for evaluating and selecting IoT-based marine vehicles"

_PeerJ Computer Science, doi:10.7717/peerj-cs.2308_

## Round 0.1 · original submission · Major Revisions

Based on the reviewer comments, the manuscript must be revised.

**Language Note:** The review process has identified that the English language must be improved. PeerJ can provide language editing services - please contact us at [email protected] for pricing (be sure to provide your manuscript number and title). Alternatively, you should make your own arrangements to improve the language quality and provide details in your response letter. – PeerJ Staff

Reviewer 1 ·

Basic reporting

The manuscript entitled “Optimizing marine vehicles industry: A hybrid AHP and ARAS approach for evaluating and selecting IoT-based marine vehicles” has been investigated in detail. The paper proposes an integrated Multi-criteria Decision-Making Analysis (MCDA) paradigm for evaluating IoT-based maritime vehicles in the automotive sector. It combines the Additive Ratio Assessment (ARAS) and Analytic Hierarchy Process (AHP) methods to assess performance and authenticity criteria. There are some points that need further clarification and improvement:
1) The paper fails to contribute significantly to existing literature, merely proposing a combination of known methodologies (ARAS and AHP) for decision-making in IoT-based maritime vehicles. This lacks innovation and fails to justify the need for the proposed approach.
2) The paper lacks clarity in its methodology description. It presents a disjointed integration of ARAS and AHP without adequately explaining how they synergize or why this integration is necessary.

Experimental design

The theoretical rationale behind the proposed integrated approach is poorly explained. There is a lack of substantive reasoning as to why ARAS is introduced to overcome the limitations of AHP.

While the paper claims empirical validation, the methodology and results of this validation are not adequately detailed. The lack of transparency in the validation process raises questions about the reliability and robustness of the proposed framework.

Validity of the findings

The study's focus on the automotive sector restricts its generalizability. The applicability of the proposed approach to other sectors or domains remains unexplored, limiting its broader impact.

“Discussion” section should be edited in a more highlighting, argumentative way. The author should analysis the reason why the tested results is achieved.

The paper falls short of providing a convincing argument for its proposed methodology. It lacks novelty, methodological rigor, theoretical justification, and robust empirical validation. Without substantial revisions addressing these concerns, the paper does not meet the standards for publication.

Additional comments

The manuscript entitled “Optimizing marine vehicles industry: A hybrid AHP and ARAS approach for evaluating and selecting IoT-based marine vehicles” has been investigated in detail. The paper proposes an integrated Multi-criteria Decision-Making Analysis (MCDA) paradigm for evaluating IoT-based maritime vehicles in the automotive sector. It combines the Additive Ratio Assessment (ARAS) and Analytic Hierarchy Process (AHP) methods to assess performance and authenticity criteria. There are some points that need further clarification and improvement:
1) The paper fails to contribute significantly to existing literature, merely proposing a combination of known methodologies (ARAS and AHP) for decision-making in IoT-based maritime vehicles. This lacks innovation and fails to justify the need for the proposed approach.
2) The paper lacks clarity in its methodology description. It presents a disjointed integration of ARAS and AHP without adequately explaining how they synergize or why this integration is necessary.
3) The theoretical rationale behind the proposed integrated approach is poorly explained. There is a lack of substantive reasoning as to why ARAS is introduced to overcome the limitations of AHP.
4) While the paper claims empirical validation, the methodology and results of this validation are not adequately detailed. The lack of transparency in the validation process raises questions about the reliability and robustness of the proposed framework.
5) The study's focus on the automotive sector restricts its generalizability. The applicability of the proposed approach to other sectors or domains remains unexplored, limiting its broader impact.
6) “Discussion” section should be edited in a more highlighting, argumentative way. The author should analysis the reason why the tested results is achieved.
The paper falls short of providing a convincing argument for its proposed methodology. It lacks novelty, methodological rigor, theoretical justification, and robust empirical validation. Without substantial revisions addressing these concerns, the paper does not meet the standards for publication.

Reviewer 2 ·

Basic reporting

no comment

Experimental design

no comment

Validity of the findings

no comment

Additional comments

The article is devoted to the actual problem of using smart solutions for deployment on the basis of IoT-based marine vehicles. The authors of the article developed a method of evaluation and selection of marine vehicles based on the Internet of Things, for which they took into account several factors and their mutual dependence. They believe the hybrid technique of AHP and ARAS provides decision makers with a powerful tool to assess the potential of IoT-based marine vehicles.
The authors described an integrated multi-criteria decision analysis (MCDA) paradigm that combines Additive Ratio Assessment (ARAS) and Analytic Hierarchy Process (AHP) approaches to evaluate and select IoT-based marine vehicles based on their efficiency and authenticity criteria in the transport sector. means In general, the article meets the requirements of the journal and can be recommended for publication in its current form.

Reviewer 3 ·

Basic reporting

The manuscript titled "Optimizing marine vehicles industry: A hybrid AHP and ARAS approach for evaluating and selecting IoT-based marine vehicles". Authors proposed the integrated Multi-criteria Decision-Making Analysis (MCDA) paradigm in this research that combines the Additive Ratio Assessment (ARAS) and Analytic Hierarchy Process (AHP) approaches to evaluate and choose IoT-based maritime vehicles based on their performance- and authenticity-related criteria in the vehicle sector.


Generally speaking, the paper is structured, organized and very well written but however the following minor corrections are suggested.

Experimental design

Methodology is badly written. rewrite it and move results to Results and discussion section

I would recommend to incorporate "Results and Discussion" section after Methodology it will contain all results with explanation.

Validity of the findings

The problem statement should be provided precisely.

Additional comments

1. Acronyms should not be there in paper title (like AHP and ARAS). Use full form

2. Avoid using word "We" and "Our" in the manuscript.

3. At the end of introduction section, Paper sections should be described as given below

"This paper comprises of five (5) sections. The first section covers the introduction of this research study. Related work with respect to research has been explained in Section 2...."

4. The manuscript should include more future directions / Limitations in Conlusion section.

5. All tables that you have presented in all steps of AHP APPROACH, should be captioned.

6. The punctuation and grammar of the manuscript should be improved considerably.

---

## Round 0.2 · Major Revisions

Dear authors,

Based on the comments of Reviewer 1, you have not adequately addressed their original comments. Please revise accordingly and provide a rebuttal letter detailing how you addressed their concerns.

Reviewer 1 ·

Basic reporting

I have thoroughly reviewed the revised manuscript, but I found that the points I previously mentioned were not comprehensively addressed. Therefore, I do not recommend the publication of the paper in its current form.

Experimental design

I have thoroughly reviewed the revised manuscript, but I found that the points I previously mentioned were not comprehensively addressed. Therefore, I do not recommend the publication of the paper in its current form.

Validity of the findings

I have thoroughly reviewed the revised manuscript, but I found that the points I previously mentioned were not comprehensively addressed. Therefore, I do not recommend the publication of the paper in its current form.

Reviewer 3 ·

Basic reporting

Problem Statement should be with "Problem Statement" heading after Literature Review section. Authors have incorporated in introduction section.

Future directions should be in separate paragraph in CONCLUSION section.

Experimental design

incorporate or results in "RESULTS AND DISCUSSION" section. why you have incorporated all results after references.

Validity of the findings

No Comment

Additional comments

Authors have incorporated all suggestions very well except some suggestions which i have mentioned in "BASIC REPORTING" and "EXPERIMENTAL DESIGN"

---

## Round 0.3 · accepted · Accept

Based on the reviewer comments, the manuscript can be accepted.

Reviewer 1 ·

Basic reporting

My comments have been addressed. It is acceptable in the present form.

Experimental design

My comments have been addressed. It is acceptable in the present form.

Validity of the findings

My comments have been addressed. It is acceptable in the present form.